# LATS1 and LATS2 suppress breast cancer progression by maintaining cell identity and metabolic state

Noa Furth[1], Ioannis S Pateras[2], Ron Rotkopf[3], Vassiliki Vlachou[2], Irina Rivkin[1], Ina Schmitt[1], Deborah Bakaev[1], Anat Gershoni[1], Elena Ainbinder[3], Dena Leshkowitz[3], Randy L Johnson[4], Vassilis G Gorgoulis[2,5,6], Moshe Oren[1], Yael Aylon[1]

Deregulated activity of LArge Tumor Suppressor (LATS) tumor suppressors has broad implications on cellular and tissue homeostasis. We examined the consequences of down-regulation of either LATS1 or LATS2 in breast cancer. Consistent with their proposed tumor suppressive roles, expression of both paralogs was significantly down-regulated in human breast cancer, and loss of either paralog accelerated mammary tumorigenesis in mice. However, each paralog had a distinct impact on breast cancer. Thus, LATS2 depletion in luminal B tumors resulted in metabolic rewiring, with increased glycolysis and reduced peroxisome proliferator-activated receptor γ (PPARγ) signaling. Furthermore, pharmacological activation of PPARγ elicited LATS2-dependent death in luminal B-derived cells. In contrast, LATS1 depletion augmented cancer cell plasticity, skewing luminal B tumors towards increased expression of basal-like features, in association with increased resistance to hormone therapy. Hence, these two closely related paralogs play distinct roles in protection against breast cancer; tumors with reduced expression of either LATS1 or LATS2 may rewire signaling networks differently and thus respond differently to anticancer treatments.

## Introduction

Breast cancer is a heterogeneous disease, with a wide spectrum of clinical, pathological, and prognostic subtypes. Although for many decades breast cancer classification was solely based on histology, nowadays studies integrating gene expression and histology are in the forefront of research (Perou et al, 2000; Sørlie et al, 2001; Curtis et al, 2012; The Cancer Genome Atlas Network, 2012), aiming to better classify and predict the clinical outcome of different tumors. Nevertheless, there remains a strong need to expand our knowledge about molecular pathways that contribute to breast cancer progression and response to therapy.

Breast cancer is broadly categorized into subtypes, with luminal and basal-like being the most common. Luminal tumors (consisting of luminal A and luminal B) express hormone receptors, underpinning their response to hormone therapies such as tamoxifen (Osborne, 1998). Luminal B (lumB) tumors tend to be of higher grade and convey a significantly worse prognosis than luminal A (lumA) tumors (Sørlie et al, 2001; Tran & Bedard, 2011; Haque et al, 2012). In contrast, basal-like tumors are mostly estrogen receptor (ER), progesterone receptor (PR), and HER2 negative (triple negative breast cancer [TNBC]) (Perou et al, 2000; Brenton et al, 2005). Importantly, decreased ER expression in lumB breast cancer has been associated with tumor recurrence and TNBC-like resistance to hormone therapy (Viale et al, 2007; Li et al, 2016). Moreover, transition of luminal tumors to invasive behavior, critical for metastasis, involves the phenotypic conversion of a luminal subpopulation to basal-like cells (Cheung et al, 2013), advocating against a rigorous division between luminal and basal-like tumors. Because of this plasticity, resistance to hormone therapy, associated with disease progression, remains a substantial therapeutic challenge.

In recent years, the LATS1 and LATS2 (LArge Tumor Suppressor [LATS]) Hippo pathway kinases have become the focus of intense research (Furth & Aylon, 2017). Classically, LATS1 and LATS2 are viewed as redundant paralogs that phosphorylate and inactivate the transcriptional cofactors YAP and TAZ (Moroishi et al, 2015). Both LATS are down-regulated in human breast cancer (Furth et al, 2015), and both have recently been implicated in modulating ER protein stability (Britschgi et al, 2017). Yet, evidence of distinct functions and differential impacts of the two paralogs is accumulating (Furth & Aylon, 2017). For instance, Lats1 knockout mice are highly sensitive to carcinogens and display pituitary dysfunction (St John et al, 1999), whereas conditional Lats2 knockout results in metabolic defects, such as fatty liver disease (Aylon et al, 2016).

[1]Department of Molecular Cell Biology, Weizmann Institute of Science, Rehovot, Israel   [2]Laboratory of Histology and Embryology Medical School, University of Athens, Athens, Greece   [3]Department of Life Sciences Core Facilities, Faculty of Biochemistry, Weizmann Institute of Science, Rehovot, Israel   [4]Department of Cancer Biology, University of Texas MD Anderson Cancer Center, Houston, TX, USA   [5]Biomedical Research Foundation of the Academy of Athens, Athens, Greece   [6]Faculty of Biology, Medicine and Health, University of Manchester, Manchester Academic Health Science Centre, Manchester, UK

Correspondence: yael.aylon@weizmann.ac.il; moshe.oren@weizmann.ac.il
Noa Furth's present address is Department of Biological Regulation, Faculty of Biology, Weizmann Institute of Science, Rehovot, Israel
Ina Schmitt's present address is Systems Biology of Signal Transduction, DKFZ, Heidelberg, Germany

Metabolic control is key to tumor suppression, reflecting the need of tumor cells to adapt their metabolism to support rapid growth. ER+ tumors often have increased fatty acid transport and elevated levels of short- and medium-chain fatty acids (Tang et al, 2014), which may affect their metabolic state, in part by regulating the activity of the nuclear peroxisome proliferator-activated receptor γ (PPARγ [Liberato et al, 2012]). This suggests a key role for PPARγ in luminal breast cancer (Zhou et al, 2009). Activation of PPARγ alters the expression of a large set of target genes, affecting adipogenesis, lipid metabolism, inflammation, and metabolic homeostasis (El Akoum, 2014). Furthermore, PPARγ activation can exert antiproliferative effects in a variety of cancer types, including breast cancer (Kersten et al, 2000; Fenner & Elstner, 2005).

Here, we show that a LATS2-associated gene expression pattern is specifically down-regulated in lumB breast cancer. Deletion of *Lats2* in the mouse mammary gland results in increased lumB tumorigenesis and metabolic rewiring of the tumor cells. Conversely, LATS2 stimulates PPARγ signaling and promotes death of lumB-derived cells. In contrast, deletion of *Lats1* reprograms lumB tumors towards basal-like characteristics. Concordantly, low LATS1 correlates with increased resistance to hormone therapy (tamoxifen). Thus, each LATS paralog exerts distinct tumor suppressive effects in the context of breast cancer, in a subtype-specific manner.

## Results

To gain insight into the impact of LATS1 and LATS2 deregulation on breast cancer, we examined the correlation between the expression levels of *LATS1* and *LATS2* in human breast cancer samples (TCGA-BRCA dataset). Although there was an overall positive correlation between the two paralogs, a subset of tumors displayed selective down-regulation of *LATS2* mRNA while retaining relatively high *LATS1* mRNA (*LATS2*L quadrant, Fig 1A). By comparing gene expression patterns between *LATS2*L and LATS^H tumors, we generated a signature of genes most down-regulated selectively in *LATS2*L tumors (*LATS2*L signature). Comparative analysis of breast cancer subtypes revealed strong down-regulation of the *LATS2*L signature particularly in lumB tumors (Fig 1B). Correspondingly, *LATS2* mRNA itself was significantly lower in lumB tumors, compared with other subtypes (Figs 1C and S1A). Importantly, decreased expression of the *LATS2*L signature was correlated with reduced survival of luminal breast cancer patients (Fig 1D); furthermore, low *LATS2* mRNA was associated with decreased probability of relapse-free survival among lumB patients (Fig S1B). Together, these observations suggest that LATS2 is a tumor suppressor in lumB breast cancer.

Mice harboring mammary gland-specific expression of the polyomavirus middle T antigen (MMTV-PyMT) develop breast tumors that recapitulate the progression of human ER+ cancer and resemble lumB tumors (Maglione et al, 2001; Herschkowitz et al, 2007; Cai et al, 2017). Hence, to explore more directly the role of LATS2 in lumB cancer, we generated MMTV-PyMT mice with mammary-specific deletion of *Lats2* (*Lats2*-CKO PyMT; Fig S2A–C). Compared with nondeleted littermate controls (WT-PyMT), deletion of *Lats2* significantly augmented mammary tumor burden (Fig 2A), formally validating the tumor suppressive function of LATS2 in mammary tumors.

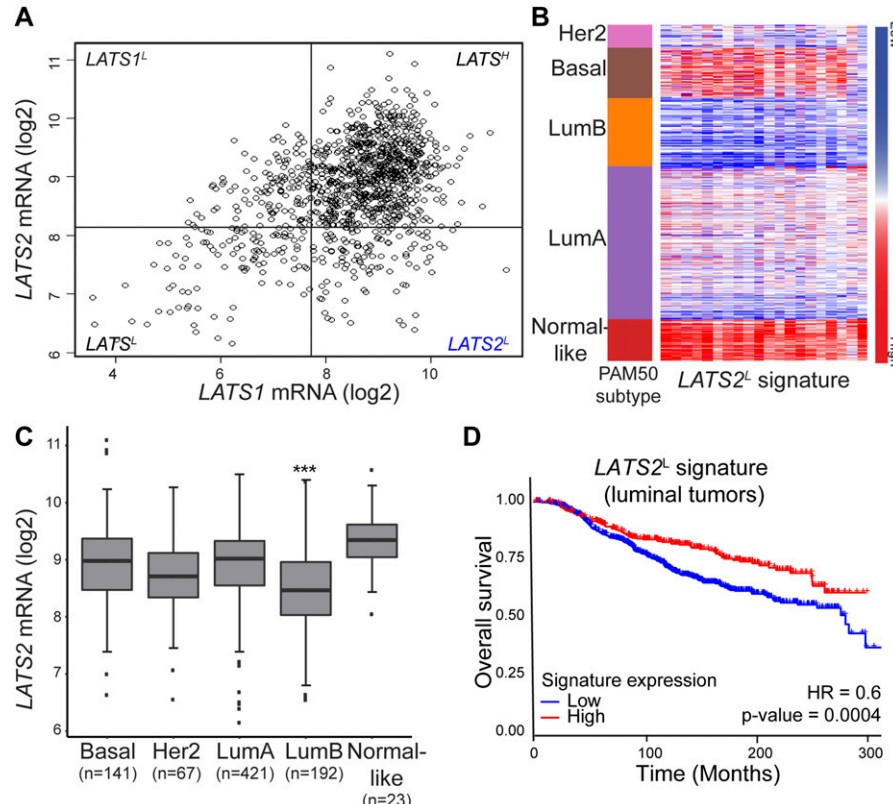

**Figure 1. LATS2-associated gene expression pattern is down-regulated specifically in lumB breast tumors. (A)** Scatter plot of *LATS1* and *LATS2* expression levels in breast cancer tumors (TCGA-BRCA dataset). Pearson's correlation coefficient 0.44. A cutoff of the 20% of tumors expressing the lowest levels of each LATS gene was used to divide the tumors into three groups: *LATS1*L, *LATS2*L, and LATS^L, compared with LATS^H. **(B)** Heatmap depicting the expression levels of a 20-gene *LATS2*L signature (the 20 most down-regulated genes exclusively in *LATS2*L tumors, compared with LATS^H tumors; see Table S4 and the Materials and Methods section) in breast tumors (TCGA-BRCA) sorted according to PAM50 subtype classification. **(C)** Distribution of *LATS2* mRNA expression levels in different breast cancer subtypes (PAM50, TCGA-BRCA); ***P-value < 0.001, t test comparing lumB tumors with all other subtypes. Number of tumors of each subtype is indicated at the bottom. **(D)** Kaplan-Meier analysis of survival probability of luminal breast cancer patients (METABRIC dataset, n = 1139; Cox proportional hazards model) divided according to expression levels of the *LATS2*L signature (as in B).

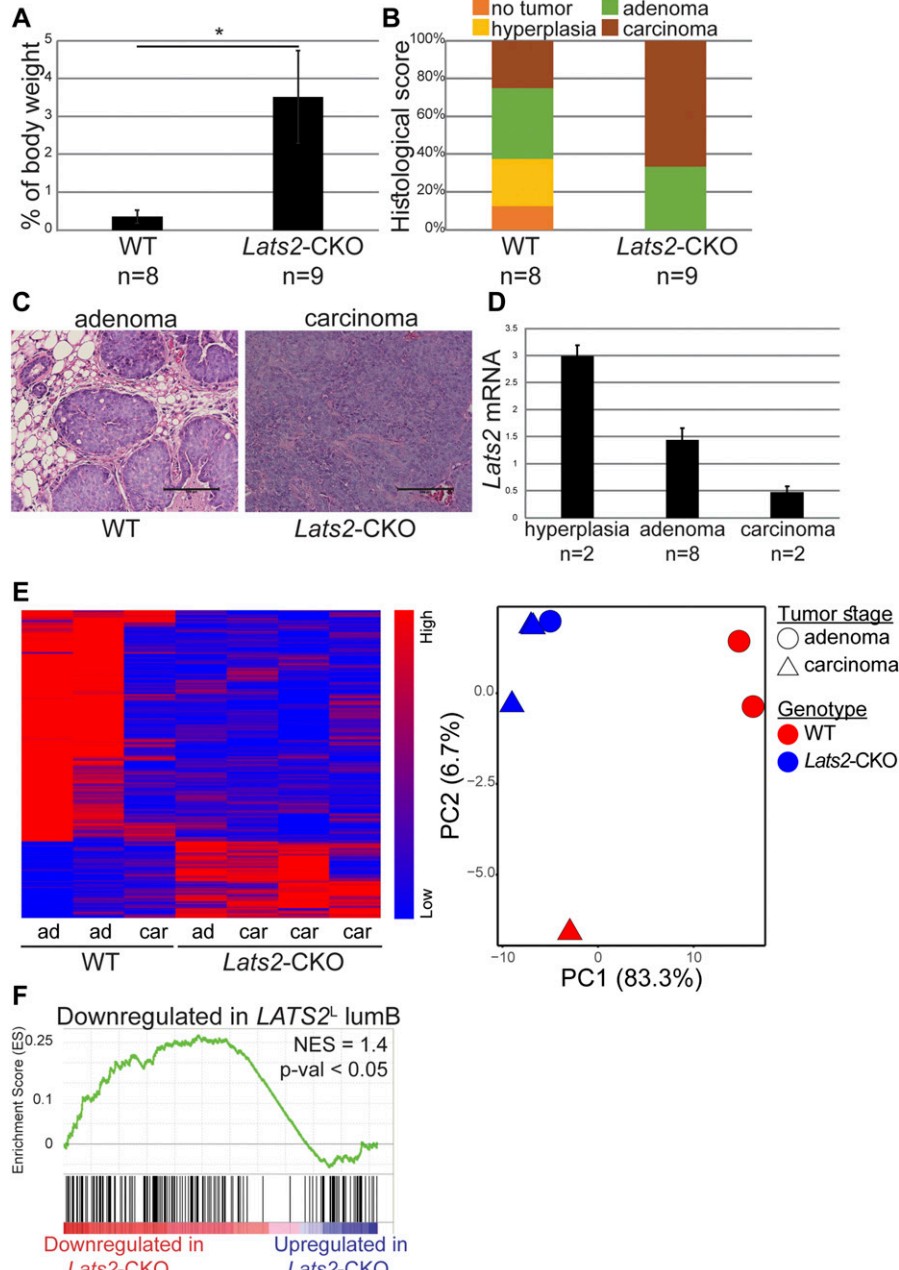

**Figure 2. LATS2 is a tumor suppressor in a mouse lumB breast cancer model.**
**(A)** Relative total tumor weight, as percentage of total body weight of 3-mo-old *Lats2*-CKO PyMT and WT-PyMT littermate controls (n = number of mice); mean ± SEM; * *P*-value < 0.05. **(B)** Three mammary glands (see the Materials and Methods section) from *Lats2*-CKO PyMT and WT-PyMT littermate control mice were histologically scored. The most advanced pathological lesion from each mouse was tallied. **(C)** Representative H&E-stained sections from WT-PyMT (adenoma/MIN) and *Lats2*-CKO PyMT (carcinoma); scale bar = 500 μm. **(D)** Expression levels of *Lats2* mRNA in WT-PyMT tumors of different histological stages, analyzed by RT-qPCR; mean ± SEM. **(E)** Left panel: Heatmap representing hierarchical clustering of global expression patterns of tumors from *Lats2*-CKO PyMT (n = 4) and WT-PyMT littermate controls (n = 3). Each tumor was taken from a different mouse. Standardized rld values are shown for differentially expressed genes (*P*-value < 0.05, n = 1131); ad = adenoma/MIN, car = carcinoma. Right panel: PCA of the most differentially expressed genes between *Lats2*-CKO PyMT and WT-PyMT tumors (adj*P*-value < 0.05), deduced from RNA-seq analysis. **(F)** GSEA of 1,131 genes ranked by fold change (red to blue gradient) between WT-PyMT and *Lats2*-CKO PyMT tumors (*P*-value < 0.05) and compared with genes down-regulated in *LATS2*[L] (versus *LATS*[H]) human lumB tumors (vertical black lines).

Importantly, by 3 mo of age, WT-PyMT mice displayed mainly adenoma/mammary intraepithelial neoplasia (MIN, [Lin et al, 2003]) and benign hyperplasia, or even no detectable pathology at all. In contrast, most of the *Lats2*-CKO PyMT mice displayed full-blown carcinomas (Figs 2B and C, and S2D). Interestingly, in line with its tumor suppressive role, *Lats2* expression declined progressively as WT-PyMT tumors became more aggressive (Fig 2D).

To further elucidate the means by which LATS2 might exert its breast tumor suppressive activity, we performed RNA sequencing (RNA-seq) analysis of WT- and *Lats2*-CKO PyMT tumors. WT-PyMT adenoma/MIN displayed a distinct transcriptional profile compared with WT-PyMT carcinoma (Fig 2E, left panel: lanes 1, 2, 3 and

right panel: PCA). Interestingly, the transcription pattern of the *Lats2*-CKO PyMT adenoma/MIN displayed remarkable resemblance to that of the WT-PyMT carcinoma, suggesting that deletion of *Lats2* facilitates a carcinoma-like gene expression pattern even at early stages of tumorigenesis. Importantly, gene set enrichment analysis (GSEA) indicated that gene expression changes in *Lats2*-CKO PyMT tumors correlated with the transcription profile of human *LATS2*[L] lumB tumors (Fig 2F), confirming the similarity of this mouse model to human lumB cancer. Moreover, genes commonly down-regulated in *LATS2*[L] human lumB and *Lats2*-CKO PyMT tumors were also found significantly associated with worse outcome in luminal breast cancer (Fig S2E). Overall, these findings demonstrate that deletion of

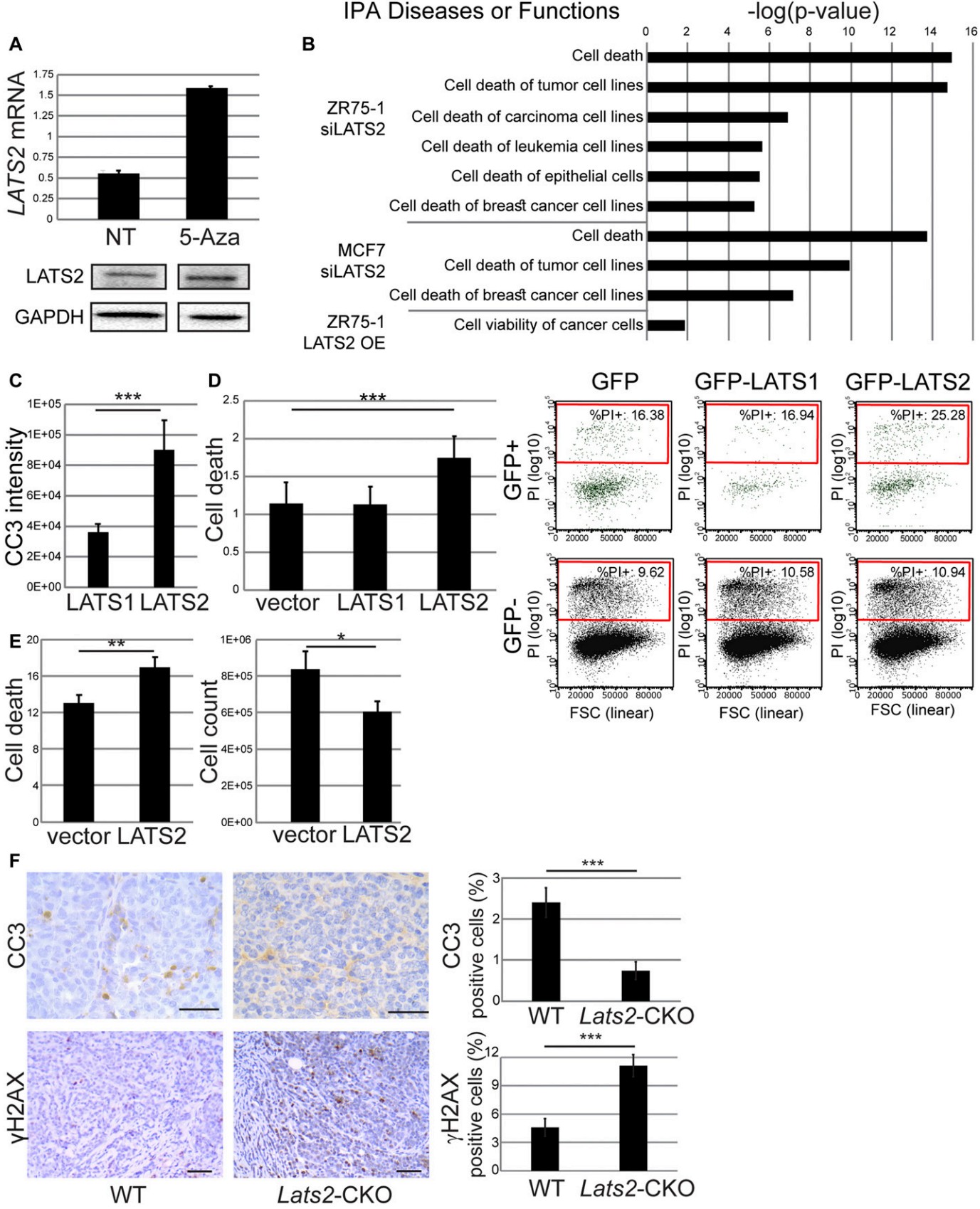

*Lats2* facilitates PyMT-driven tumorigenesis, further supporting the role of LATS2 as a tumor suppressor in human lumB breast cancer.

To further explore the impact of LATS2 down-regulation on human lumB cancer, we used our *LATS2*[L] signature to probe breast cancer cell lines in the Cancer Cell Line Encyclopedia (Barretina et al, 2012). Consistent with the human patient data, the *LATS2*[L] signature was down-regulated in cell lines derived from luminal cancers, including MCF7 and ZR75-1 (Fig S3A, Holliday & Speirs, 2011). Decreased *LATS2* expression in breast cancer has been associated with promoter hypermethylation (Takahashi et al, 2005). Specifically, the CpG island surrounding the *LATS2* transcription start site displayed increased methylation in *LATS2*[L] lumB tumors relative to *LATS2*[H] tumors (Fig S3B). Indeed, inhibition of DNA methylation by 5-aza-2'deoxycytidine (5-Aza) treatment increased the levels of *LATS2* mRNA and protein in lumB-derived ZR75-1 cells (Fig 3A), suggesting that lumB breast cancers might undergo selective pressure to silence *LATS2* expression.

Next, we assessed the transcriptional impact of LATS2 modulation in cell lines derived from human luminal cancers. To that end, we performed RNA-seq analysis on ZR75-1 and MCF7 cells subjected to siRNA-mediated silencing of *LATS2*, as well as ZR75-1 cells over-expressing exogenous *LATS2*. Modulation of LATS2 levels in these cells resulted in expression changes relating to different signaling pathways (Table S1A). Unexpectedly, YAP/TAZ gene signatures were not among the most significantly enriched, rather, in all cases, LATS2 modulation elicited expression changes in genes related to cell death (Fig 3B and Table S1B). Concordantly, transient or stable overexpression of LATS2 (Fig S3C) in ZR75-1 cells elicited increased apoptosis, assessed by cleaved caspase 3 (CC3), and reduced cell viability, assessed by pro-pidium iodide (PI) exclusion (Fig 3C–E); this was not seen with LATS1. In contrast, LATS2 overexpression did not reduce the viability of MDA-MB-468 cells, derived from basal-like breast cancer (Fig 3D), suggesting a luminal cancer-specific proapoptotic role of LATS2. In line with the increased death upon LATS2 overexpression in human luminal cancer cells, mouse *Lats2*-CKO PyMT tumors displayed reduced numbers of CC3-positive cells, relative to WT-PyMT tumors (Fig 3F). Interestingly, this was concomitant with augmented γH2AX staining, indicative of sustained DNA damage (Fig 3F). Notably, inability to clear DNA-damaged cells has been associated with aggressive clinicopatholog-ical features and poor patient outcome (Yang et al, 2017). Together, this implies that LATS2 might suppress tumorigenesis in part by promoting the elimination of damaged cells.

To further elucidate the underpinnings of LATS2-mediated tumor suppression, we analyzed the functional differences between the transcriptional profiles of *Lats2*-CKO PyMT and WT-PyMT tumors (see Fig 2E). Enrichment analysis (Fig 4A) revealed pronounced deregulation of numerous metabolic terms, including "regulation of lipolysis," "TCA cycle," and "glycolysis/gluconeogenesis," with "PPAR signaling" being the most significantly altered. Likewise, the top predicted upstream regulator of LATS2-dependent genes in this comparison was PPARγ (ingenuity activation z-score 3.7; *P*-value of overlap $7.8 \times 10^{-17}$). Similar metabolism-associated changes in gene expression were seen also when LATS2 was transiently overexpressed in ZR75-1 cells or silenced by siRNA in either ZR75-1 or MCF7 cells (Table S2). In line with LATS2-dependent metabolic homeostasis, stable overexpression of LATS2 in ZR75-1 cells (ZR75-1/LATS2) augmented cell death under regular culture conditions but dampened the further increase in cell death upon glucose deprivation when compared with control ZR75-1 cells (ZR75-1/vector, Fig 4B). These observations sug-gested that the inherently low levels of LATS2 in ZR75-1 cells might render them more dependent on extracellular glucose to maintain high aerobic glycolysis ("Warburg effect"). Furthermore, high LATS2 may favor oxidative phosphorylation, conferring less dependence on supplemented glucose. Indeed, Seahorse metabolic analysis con-firmed that the glycolytic rate was diminished in ZR75-1/LATS2 relative to control ZR75-1/vector cells, whereas respiratory capacity was augmented (Fig 4C). In agreement, cell lines derived from *Lats2*-CKO PyMT tumors (Fig S4A and B) mirrored these observations, per-forming less respiration than WT-PyMT cells (Fig 4D). Overall, these results suggest that down-regulation of LATS2 may facilitate the survival of cells that have undergone metabolic rewiring towards a Warburg-like profile, favoring glycolysis over oxidative respiration.

Dampening of PPARγ activity might reflect an energetic necessity of highly glycolytic LATS2-depleted cancer cells. Indeed, genes commonly down-regulated in both *Lats2*-CKO PyMT tumors and *LATS2*[L] lumB human breast cancers were significantly enriched for "PPAR signaling" (Fig 5A), confirming the link between LATS2 and PPARγ activity in both mouse and human tumors. Decreased ex-pression of *Pparg* (subject to positive autoregulation [Wakabayashi et al, 2009]) and its transcriptional target *Plin1* was confirmed by RT-qPCR in an expanded number of *Lats2*-CKO PyMT tumors (Fig 5B). Furthermore, *Pparg* and its target genes *Plin1* and *Lpl* were down-regulated also in cultured *Lats2*-CKO PyMT cells (Fig 5C), implicating a cell-autonomous regulatory interaction between LATS2 and PPARγ. Of note, stable expression of human LATS2 in the *Lats2*-CKO PyMT cells (*Lats2*-CKO/LATS2) resulted in increased expression of *Pparg* and its target genes (Fig 5C). Interestingly, in LATS2-overexpressing ZR75-1 cells, augmented PPARγ was selectively

---

**Figure 3. LATS2 promotes death of lumB cells.**
**(A)** ZR75-1 cells were treated with 1 μM 5-aza-2'-deoxycytidine (5-Aza) for 4 d. Upper panel: RT-qPCR analysis of *LATS2* mRNA; mean ± SD of two technical replicates. Lower panel: Western blot analysis of LATS2 protein. **(B)** Functional enrichment of cell viability-related terms for differentially expressed genes in luminal cancer cell lines with transient silencing (siLATS2) or stable overexpression (OE) of LATS2, compared with controls. Ingenuity Pathway Analysis (IPA); Diseases or Functions annotations. **(C)** ZR75-1 cells were transiently transfected with *GFP-LATS1* or *GFP-LATS2*, stained with anti-cleaved caspase 3 (CC3) antibody after 48 h, and subjected to imaging flow cytometry (ImageStreamX). Only cells with intact nucleus and positive GFP signal were analyzed; mean ± SEM of staining intensity; ***P*-value < 0.001. **(D)** ZR75-1 cells were transfected with *GFP-LATS1* or *GFP-LATS2* or *GFP* only (vector). 72 h later, cell viability was assessed by PI exclusion followed by flow cytometry analysis. "Cell death" represents the ratio between the percentages of dead cells (PI[+]) in the GFP-positive population versus the GFP-negative population; mean ± SEM of three independent experiments; ***P*-value < 0.001. Representative FACS results are shown on the right; in each case, upper and lower panels represent the GFP-positive and GFP-negative subpopulation, respectively, of the same transfected culture. **(E)** ZR75-1 cells stably transduced with *MYC-LATS2* plasmid or control vector were subjected to PI exclusion analysis followed by flow cytometry. Left panel: percentage of dead (PI[+]) cells. Right panel: cell count of live (PI[−]) cells; mean ± SEM of three independent experiments; *P*-value < 0.05, **P*-value < 0.01. **(F)** Histological samples from carcinomas of *Lats2*-CKO PyMT and WT-PyMT littermate controls were immunostained for γH2AX and cleaved caspase 3 (CC3, left panel, scale bar = 100 μm). Right panel: mean ± SEM of percentage of positive cells, based on at least nine sections of each genotype; ***P*-value < 0.001.

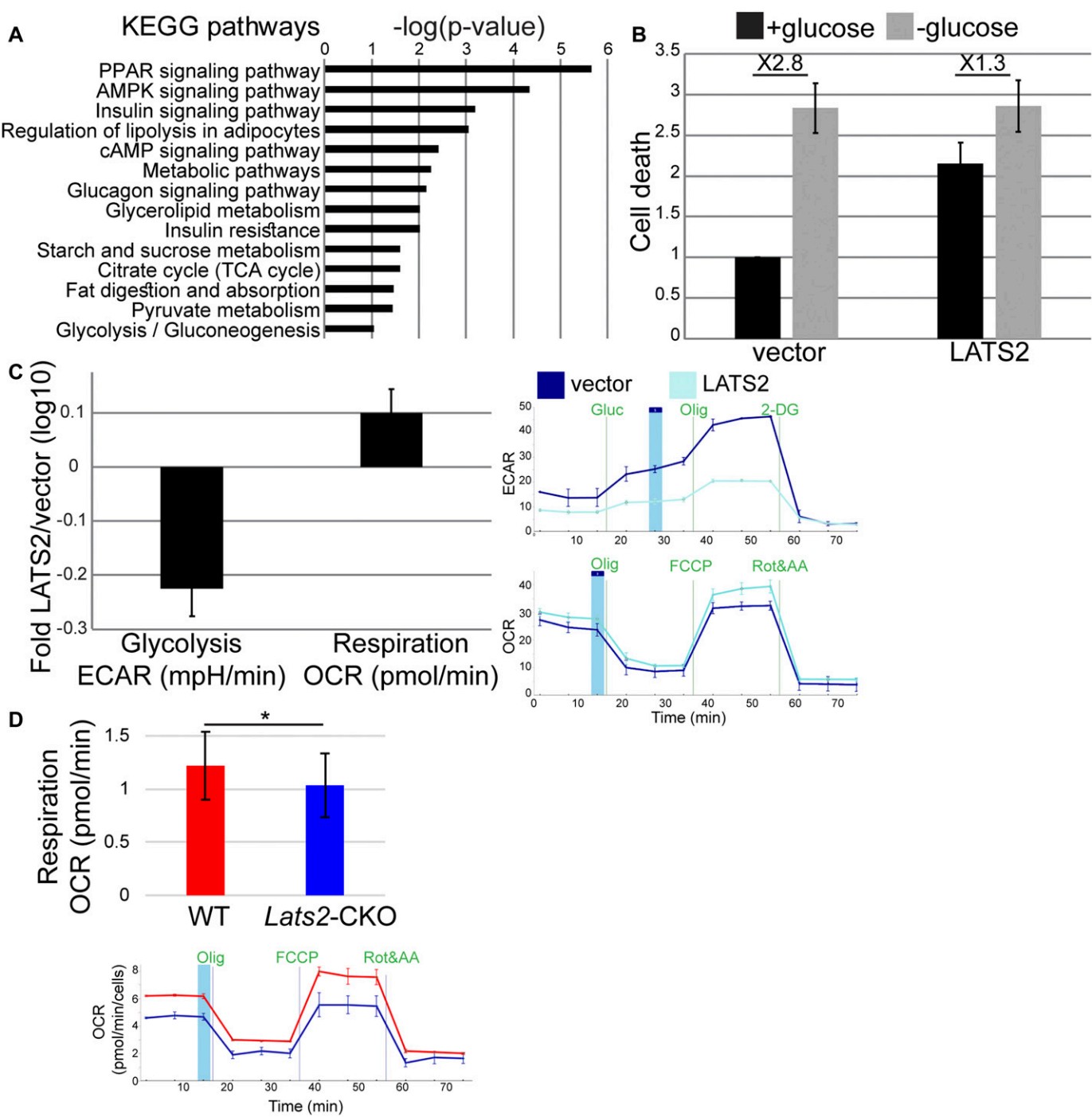

**Figure 4. LATS2 augments oxidative phosphorylation.**
**(A)** KEGG pathways enrichment analysis for genes differentially regulated in *Lats2*-CKO PyMT compared with WT-PyMT tumors (adj*P*-value < 0.05, n = 114). **(B)** ZR75-1 cells stably transfected with an *MYC-LATS2* plasmid (LATS2) or with control vector were cultured with or without glucose for 3 d. Cell death was measured by PI exclusion followed by FACS analysis. Values from each experiment were normalized to % PI⁺ of control cells grown in glucose-containing medium; mean ± SEM of three independent experiments. **(C)** Extracellular acidification rate (ECAR, indicative of glycolysis) and oxygen consumption rate (OCR, indicative of respiration) of ZR75-1 cells stably expressing *MYC-LATS2*, relative to vector control cells, determined by Seahorse; mean ± SEM of five independent experiments. Representative tracks of Seahorse measurements are shown on the right. **(D)** Respiration measured by OCR in WT-PyMT and *Lats2*-CKO PyMT tumor-derived cell lines; mean ± SEM in log 10 scale of three independent experiments; *P*-value < 0.05. Bottom: representative Seahorse track.

associated with nuclear localization of LATS2 (Fig 5D), suggesting that a nuclear function of LATS2 is responsible for promoting PPARγ expression.

Analysis of human luminal tumors revealed that *LATS2* and *PPARG* mRNA levels are positively correlated (Fig 5E, left); moreover, *LATS2*ᴸ lumB tumors displayed reduced PPARγ signaling (Fig 5E,

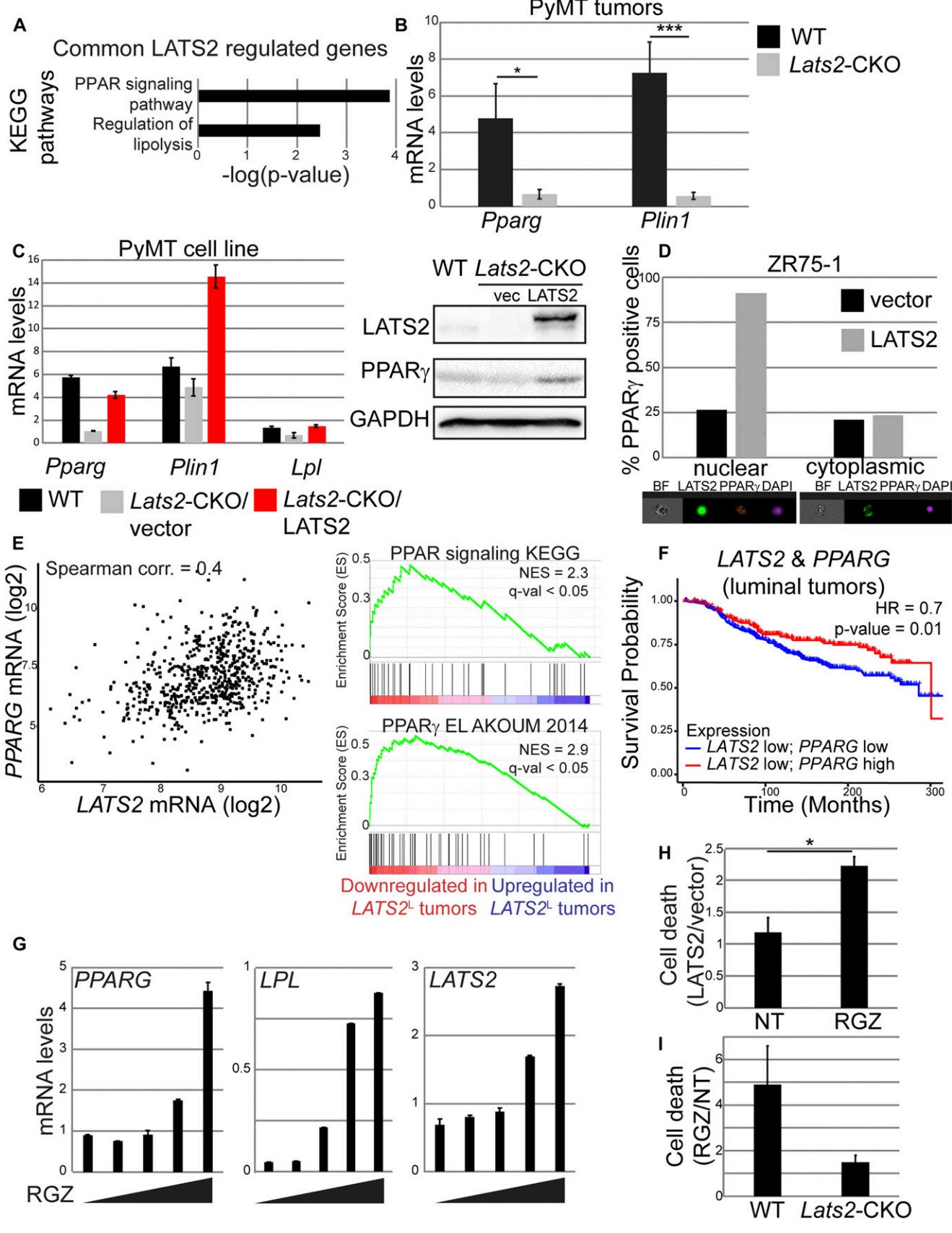

right). Of note, similar to *LATS2* overexpression in ZR75-1 cells (see Fig 3A), 5-Aza treatment increased the expression of both *PPARG* and *LPL* (Fig S5A), suggesting that both *LATS2* expression and PPARγ activity are epigenetically down-regulated during lumB tumorigenesis. In contrast, *PPARG* expression in luminal tumors was not significantly correlated with *LATS1* expression (Fig S5B). Interestingly, luminal breast cancer patients with low expression of both *LATS2* and *PPARG* displayed impaired overall survival, compared with patients with low *LATS2* but high *PPARG* expression (Fig 5F). Altogether, our observations support the notion that LATS2 down-regulation, which leads to reduced PPARγ signaling, may facilitate metabolic rewiring towards a more favorable energetic state, associated with more lethal tumors.

PPARγ activation has been shown to inhibit proliferation and induce cell cycle arrest in breast cancer cells (Mueller et al, 1998; Suh et al, 1999; Mehta et al, 2000). Rosiglitazone (RGZ) is an FDA-approved antidiabetic drug (Soccio et al, 2014) that selectively activates PPARγ (Willson et al, 1996). As expected, treatment of ZR75-1 cells with RGZ resulted in increased PPARγ activity, reflected by augmented expression of the PPARγ target gene *LPL* as well as of the *PPARG* mRNA itself (Fig 5G, left panel). Interestingly, activation of PPARγ also resulted in up-regulation of *LATS2* mRNA (Fig 5G, right panel), implying a positive feedback between the two. Re-expression of LATS2 conferred hypersensitivity to RGZ (Figs 5H and S5C), suggesting that high LATS2 may further sensitize these cells to excessive PPARγ activation. In agreement, partial silencing of LATS2, but not LATS1, reduced sensitivity to RGZ (Fig S5D). Moreover, deletion of *Lats2* nearly abolished the death of cultured PyMT mouse tumor cells upon RGZ treatment (Fig 5I), further confirming that sensitivity of lumB breast cancer cells to PPARγ activation is LATS2 dependent.

Next, we wished to examine the impact of *Lats1* on breast cancer development in PyMT mice. To that end, we generated MMTV-Cre PyMT mice with mammary gland-specific knockout of *Lats1* (*Lats1*-CKO PyMT) (Fig S6A–C). Similar to *Lats2* deletion, conditional knockout of *Lats1* caused a significant increase in tumor burden relative to WT-PyMT littermate controls (Fig 6A). Surprisingly, a notable portion of *Lats1*-CKO PyMT mice developed adenosquamous carcinoma (Fig 6B), a tumor type not observed in *Lats2*-CKO PyMT mice (see Fig 2B). This was particularly striking because the MMTV-PyMT model typically represents ER+ lumB cancer (Maglione et al, 2001; Herschkowitz et al, 2007; Cai et al, 2017), whereas human adenosquamous carcinoma is invariably triple negative (Geyer et al, 2017). In line with this, *Lats1*-CKO PyMT tumors displayed significantly reduced ER positivity

(Fig 6C) and increased abundance of the basal cell marker CK14 (Fig 6D). This suggests that differently from LATS2, LATS1 might be important for maintaining the expression of luminal cell markers and restricting cell plasticity. Notably, *LATS1*, but not *LATS2*, was down-regulated in human breast metaplastic carcinoma (Fig S6D), which is thought to occur via transdifferentiation of a subpopulation of cancer cells (van Deurzen et al, 2011). Metaplastic carcinomas are typically negative of hormone receptors (ER/PR) and HER-2/neu, and are histologically characterized by mixed epithelial and transdifferentiated components (Aydiner et al, 2015), thus resembling the adenosquamous carcinomas detected in the *Lats1*-CKO PyMT tumors. Altogether, this suggests that down-regulation of LATS1, but not LATS2, favors partial loss of luminal identity of tumor cells.

We next compared the global gene expression patterns of *Lats1*-CKO PyMT tumors and their WT-PyMT littermate control tumors (Fig 6E). Overall, the expression pattern changes in the mouse *Lats1*-CKO PyMT tumors agreed well with those seen in human luminal cancer cell lines (MCF7 and ZR75-1) subjected to either up- or down-modulation of LATS1 (Fig S6E), supporting the human relevance of the *Lats1*-CKO PyMT model. Of note, the altered signaling pathways pattern in *Lats1*-CKO PyMT tumors was distinct from that observed upon depletion of *Lats2*. For example, unlike in *Lats2*-CKO PyMT tumors, PPARγ signaling was only mildly deregulated in *Lats1*-CKO PyMT tumors (Fig 6F). Also, up-regulation of YAP/TAZ target genes was more apparent in *Lats1*-CKO PyMT tumors and following *LATS1* silencing in ZR75-1 cells, as compared with *Lats2*-CKO PyMT tumors (Fig S6F) or *LATS2* silencing, respectively (Fig S6G). This is consistent with canonical YAP/TAZ inhibition by LATS1 in the human breast cancer setting (Zhang et al, 2008; Cordenonsi et al, 2011; Britschgi et al, 2017). Furthermore, in agreement with the known ability of hyperactive YAP/TAZ to augment cell proliferation, *LATS1* silencing in human luminal breast cancer-derived cells yielded a transcriptional signature strongly enriched for cell cycle and mitosis-related pathways (Fig S6H).

Congruous with the changes in ER and CK14 levels in *Lats1*-CKO PyMT tumors (see Fig 6C and D), the expression pattern of *Lats1*-CKO PyMT tumors was similar to other ER-negative mouse mammary tumors (Fig 7A) and dissimilar to mature luminal cells (Fig 7B). Furthermore, it was significantly similar to that of human basal-like tumors, compared with luminal tumors (Fig 7C). In addition, we derived a basal-like expression signature from human breast cancer samples (TCGA dataset, see the Materials and Methods section) and found it to be up-regulated selectively in the *Lats1*-CKO PyMT, but not

---

**Figure 5. PPARγ signaling correlates with LATS2 in human and mouse tumors and promotes cell death in a LATS2-dependent manner.**
**(A)** KEGG pathways significantly enriched within the list of genes commonly down-regulated in *Lats2*-CKO PyMT (compared with WT-PyMT) tumors and *LATS2*^L lumB human tumors (TCGA). **(B)** RT-qPCR quantification of *Pparg* and *Plin1* expression in tumors derived from 3 mo old *Lats2*-CKO PyMT (n = 9) and WT-PyMT littermate controls (n = 6); mean ± SEM; *P-value < 0.05, ***P-value < 0.001. **(C)** Expression levels of the indicated genes (left) and proteins (right) in cultured WT PyMT and *Lats2*-CKO PyMT tumor-derived cells stably transduced with LATS2 or vector control; mean ± SD of 2 technical repeats. **(D)** ZR75-1 cells were transfected with *GFP-LATS2* or control *GFP* plasmid, stained 48 h later with anti-PPARγ and cleaved caspase 3 (CC3) antibodies, and subjected to imaging flow cytometry (ImageStreamX). Only GFP-positive cells with intact nuclei were analyzed. Cells with nuclear localization of the transfected protein were identified by similarity between GFP and DAPI staining. Percentage of cells positively stained for PPARγ in each subpopulation is presented. Representative images are shown at the bottom. BF = bright field. **(E)** Left panel: scatter plot depicting *PPARG* and *LATS2* expression in luminal tumors in the TCGA-BRCA dataset (n = 613). Right panel: GSEA of genes ranked according to expression fold change between *LATS2*^L tumors and LATS^H tumors in the TCGA lumB dataset. **(F)** Kaplan-Meier analysis of survival probability of luminal breast cancer patients (METABRIC dataset, n = 1139; Cox proportional hazards model) defined according to expression levels of *LATS2* and *PPARG*. **(G)** RT-qPCR quantification of *PPARG* and *LPL* (left) and *LATS2* (right) mRNA in ZR75-1 cells treated for 48 h with increasing concentrations of RGZ (0, 10, 25, 50, and 100 μM, respectively); mean ± SD of two technical replicates. **(H)** ZR75-1/vector (vector) and ZR75-1/LATS2 (LATS2) cells were treated with 100 μM RGZ for 48 h, followed by PI exclusion analysis; mean ± SEM of the ratio between % PI⁺ cells in each cell type, from three independent experiments. **(I)** Cell lines derived from WT-PyMT and *Lats2*-CKO PyMT tumors were treated with 100 μM RGZ for 48 h, followed by PI exclusion analysis; mean ± SEM of the ratio between % PI⁺ cells in treated versus untreated cells, from two independent experiments.

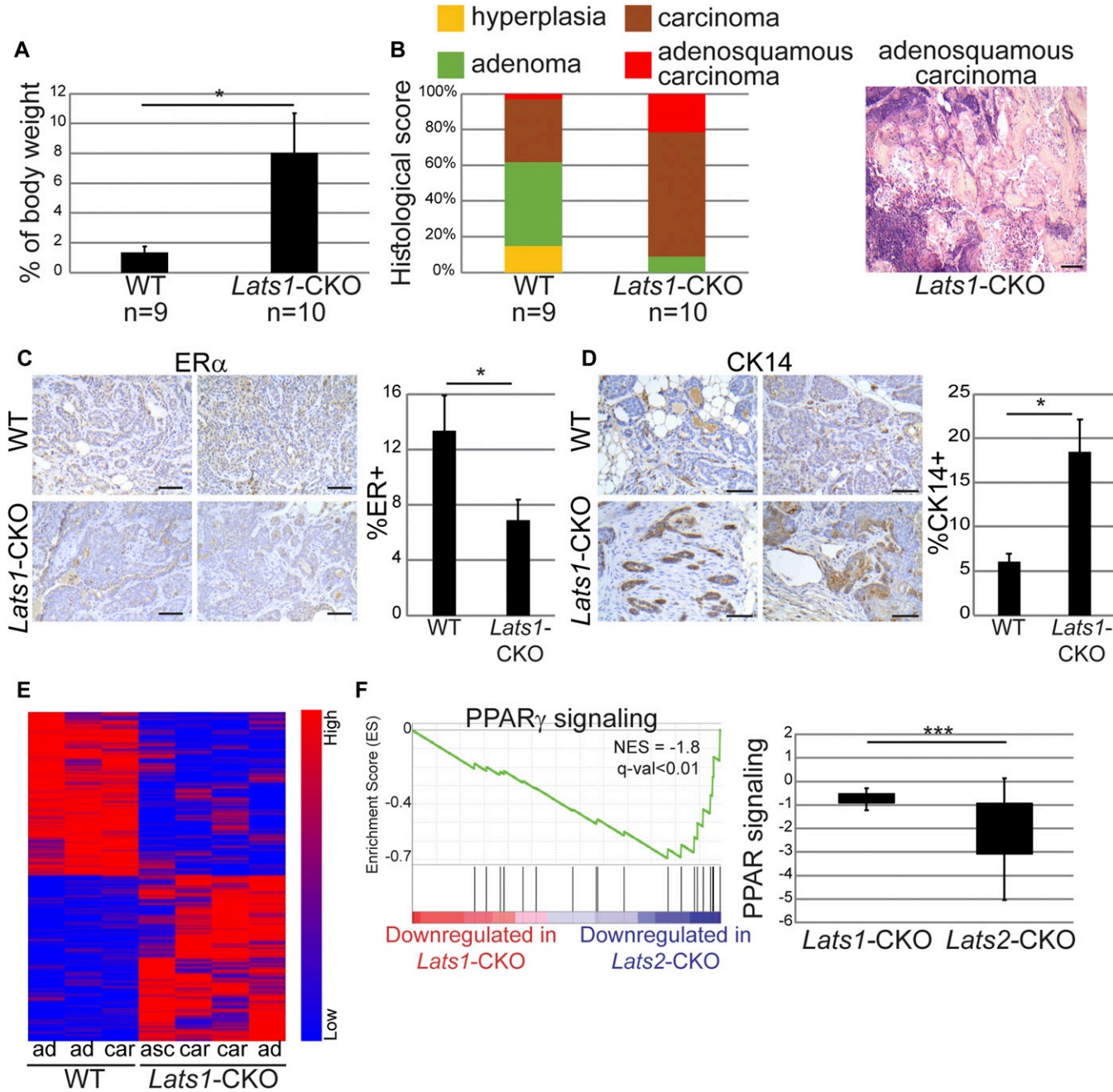

**Figure 6. Deletion of *Lats1* is phenotypically distinct from *Lats2* deletion in PyMT tumors.**
**(A)** Relative tumor weight, as percentage of total body weight, in three month old *Lats1*-CKO PyMT and WT-PyMT littermate controls (n = number of mice); mean ± SEM; *P*-value < 0.05. **(B)** Three mammary glands from *Lats1*-CKO PyMT and WT-PyMT littermate controls were histologically scored and tallied. Right panel: representative H&E-stained adenosquamous carcinoma sample from a *Lats1*-CKO PyMT tumor (scale bar = 200 μm). **(C)** Immunohistochemistry analysis of ERα protein expression in tumors from 3-mo-old *Lats1*-CKO PyMT and WT-PyMT littermate controls. Left panel: two representative tumor sections from each genotype (scale bar = 100 μm). Right panel: mean ± SEM of % ER$^+$ cells in tumors from eight *Lats1*-CKO PyMT and 6 WT-PyMT mice; *P*-value < 0.05. **(D)** Immunohistochemistry analysis of CK14, performed as in (C). Right panel depicts mean ± SEM of percentage of slide area with intense membranous CK14 staining in the invasive front of the tumor; *P*-value < 0.05. **(E)** Heatmap representing hierarchical clustering of global expression patterns of tumors from 3-mo-old *Lats1*-CKO PyMT and WT-PyMT littermate controls. Standardized rld values are shown for differentially expressed genes (*P*-value < 0.05, n = 2029). ad = adenoma/MIN, car = carcinoma, asc = adenosquamous carcinoma; **(F)** Left panel: GSEA assessing PPARγ transcriptional activity in *Lats1*-CKO PyMT versus *Lats2*-CKO PyMT tumors. Genes differentially expressed (adj*P*-value < 0.05) in either *Lats1*-CKO PyMT (compared with littermate controls) or *Lats2*-CKO PyMT (compared with littermate controls) were ranked according to fold change differences (log ratio) and compared with a gene set comprising PPARγ target genes (El Akoum, 2014). Right panel: box plot representing mean fold change (log 2) of genes comprising the PPAR pathway (KEGG database) in *Lats1*-CKO PyMT and *Lats2*-CKO PyMT, each compared with their WT-PyMT littermate controls. Only genes with *P*-value of comparison < 0.05 were included; ***P*-value < 0.001.

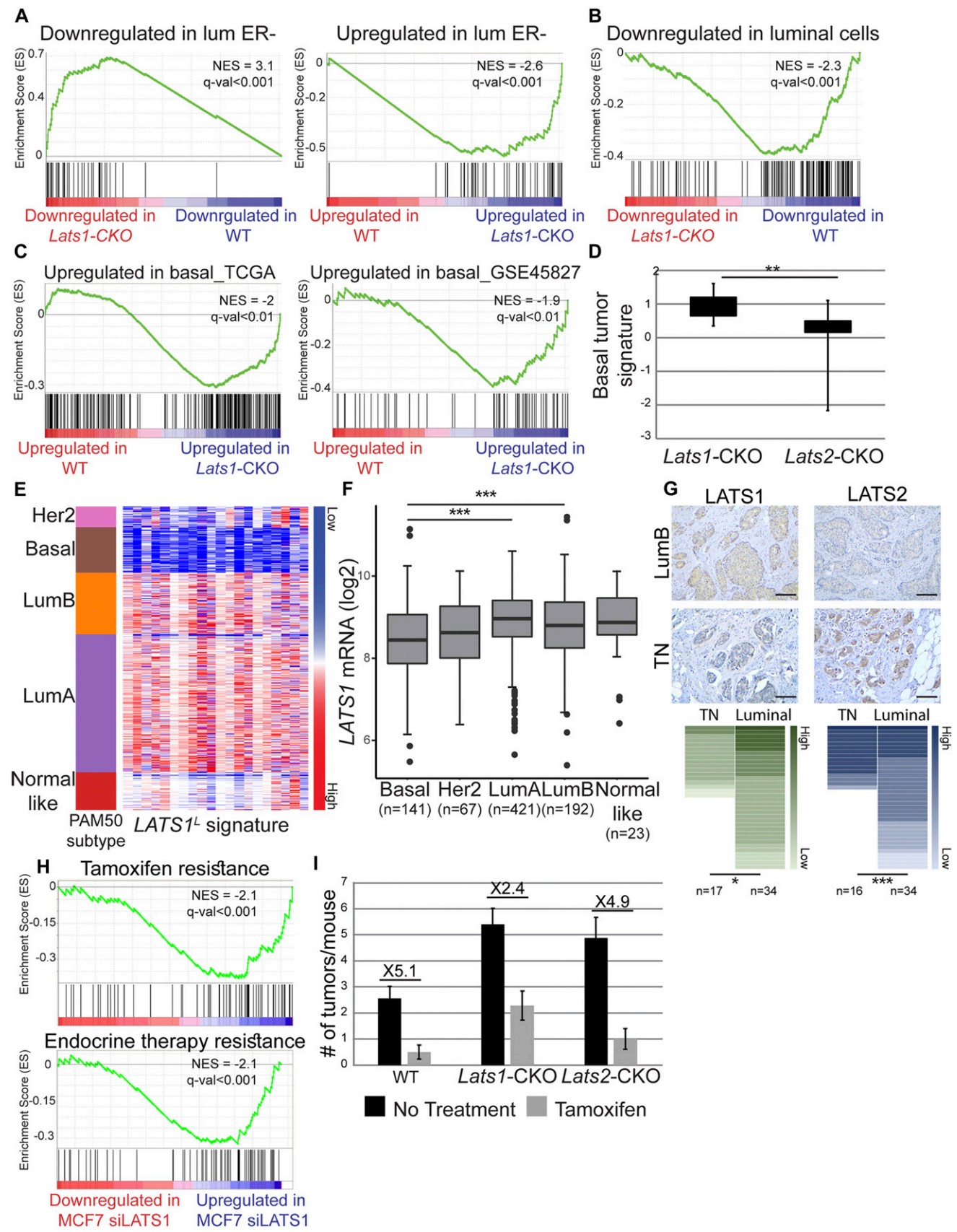

*Lats2*-CKO tumors (each compared with WT-PyMT littermate control tumors, Fig 7D and Table S3). Overall, this implies that LATS1 is important for maintaining luminal breast cancer-associated expression patterns; without it, tumors may become more basal-like, even within an otherwise lumB-predisposed in vivo setting. To further examine this possibility, we generated a *LATS1*[L] 20-gene signature, comprising genes that were selectively down-regulated in *LATS1*-low human breast tumors (see Fig 1A) and evaluated its expression in different breast cancer subtypes. Notably, this signature (Fig 7E), as well as *LATS1* mRNA itself (Fig 7F), was strongly down-regulated particularly in basal-like tumors. Furthermore, LATS1 protein levels were lower in TNBC tumors relative to luminal tumors, whereas the opposite trend was seen for LATS2 (Fig 7G).

Silencing of *LATS1* in MCF7 cells resulted in increased expression of genes associated with resistance to hormone therapy, particularly tamoxifen (Fig 7H), supporting a role for LATS1 in maintaining estrogen dependence of luminal breast cancer cells. To assess more directly whether deletion of *Lats1* might facilitate hormone therapy resistance, we monitored the response of *Lats1*-CKO PyMT tumors to tamoxifen treatment. Tamoxifen treatment retarded tumor growth in all genotypes (Fig 7I); however, *Lats1*-CKO PyMT tumors were relatively less inhibited than WT-PyMT or *Lats2*-CKO PyMT tumors. Overall, our observations suggest that the partial loss of luminal identity upon LATS1 down-regulation may promote resistance to hormone therapy, paralleling the clinical manifestations of tamoxifen-resistant human breast tumors (Kuukasjärvi et al, 1996).

## Discussion

Previous studies have demonstrated that reduced expression of LATS kinases is associated with breast cancer characteristics in vitro (Zhang et al, 2008; Furth et al, 2015; Li & Gumbiner, 2016). We provide evidence that both LATS1 and LATS2 are *bona fide* tumor suppressors in an in vivo breast cancer setting. In a lumB breast cancer mouse model, conditional deletion of either paralog increases tumorigenesis, both in magnitude and severity. Importantly, in both human and mouse breast tumors, down-regulation of each paralog is associated with a distinct breast cancer subtype and deregulation of different signaling pathways; decreased LATS1 compromises the strict maintenance of luminal cell fate and favors a drift towards a more basal-like state, whereas decreased LATS2 rewires metabolism towards reduced PPARγ activity and increased glycolysis.

Differential impact of LATS1 and LATS2 on distinct molecular signaling pathways is in line with a growing body of evidence that each paralog can operate within a functional spectrum that includes also unique, nonredundant activities (Furth & Aylon, 2017). Within this spectrum, LATS may sometimes even contribute to cancer progression, for example, by suppressing anticancer immunity (Moroishi et al, 2016). Yet, most of our current knowledge is more consistent with a tumor suppressive role of these proteins. Notably, both LATS proteins inhibit the tumor-promoting activities of YAP and TAZ (Moroishi et al, 2015). Indeed, loss of YAP has been shown to suppress the growth of PyMT-driven mammary tumors (Chen et al, 2014). Interestingly, we observed that deletion of *Lats1*, but not *Lats2*, results in a moderate increase of YAP/TAZ-associated transcriptional activity in such tumors. Nevertheless, in other contexts, LATS2 does contribute to the inhibition of YAP/TAZ-mediated phenotypes (Meng et al, 2015). Thus, the contribution of each LATS kinase to the regulation of YAP/TAZ activity is markedly context dependent.

We found that LATS2 augments PPARγ activity in vitro and in vivo. Of note, PPARγ was reported to suppress glycolysis and induce apoptosis in breast cancer cells by repressing the expression of several glycolytic enzymes (Shashni et al, 2013). Hence, the attenuated PPARγ signaling in LATS2-depleted breast cancer cells may account, at least in part, for the observed increase in glycolysis and the increased dependence on glucose for survival. Moreover, we observed that LATS2 is required for induction of luminal breast cancer cell death by the PPARγ agonist RGZ. We propose that inefficient clearance of highly glycolytic, genomically altered cells with low levels of LATS2 might represent a devious evolutionary adaptation of lumB tumors. Conversely, simultaneous activation of LATS2 and PPARγ may instigate a metabolic catastrophe, particularly in cancer cells with high energy demands, possibly by increasing mitochondrial burden (Zolezzi et al, 2013). PPARγ agonists have shown efficacy against some solid tumors in clinical trials (Demetri et al, 1999); our data suggest that LATS2 status might be a determinant of the success of such agents. More broadly, the ability of LATS2 to modulate master regulators of lipid metabolism such as PPARγ and SREBP (Aylon et al, 2016) expands the functional consequences of the deregulation of Hippo pathway components in cancer and may indicate a metabolic Achilles' heel of particular tumors.

Although PPARγ itself was not functionally enriched by depletion of hepatic *Lats2* (Aylon et al, 2016), several other metabolic pathways were commonly enriched by deletion of *Lats2* in both breast and liver (Fig S7). One such pathway is SREBF1 (also known as SREBP1), which is

---

**Figure 7. Down-regulation of LATS1 promotes the formation of tumors enriched in basal-like features.**
**(A)** Genes were ranked according to fold change between *Lats1*-CKO PyMT and WT-PyMT tumors (*P*-value < 0.05), and compared with genes differentially expressed in ER-negative PyMT tumors (GSE64453), using GSEA. **(B)** Genes were ranked as in (A) and compared with genes down-regulated in mature luminal cells relative to mammary stem cells (GSE19446), using GSEA. **(C)** Genes were ranked as in (A) and compared with genes up-regulated in human basal-like tumors relative to luminal tumors (TCGA, see the Materials and Methods section) (left panel) or genes up-regulated in human basal tumors relative to lumB (GSE45827), using GSEA. **(D)** Box plot representing mean fold change (log2) of genes comprising the basal tumor signature (see the Materials and Methods section and Table S3), in *Lats1*-CKO PyMT and *Lats2*-CKO PyMT tumors compared with their WT-PyMT littermate controls. **(E)** Heatmap depicting the expression levels of the 20 most down-regulated genes in *LATS1*[L] tumors relative to *LATS*[H] tumors (see Table S4 and the Materials and Methods section). Tumors were sorted according to the PAM50 classification. **(F)** Relative expression levels of *LATS1* in different breast cancer subtypes (PAM50 and TCGA-BRCA). Numbers of samples are indicated at the bottom. ANOVA coupled with Dunnett's test; ****P*-value < 0.01. **(G)** Immunohistochemistry (IHC) analysis of LATS1 and LATS2 in luminal and triple negative (TN) human tumors. Representative tumor sections of human lumB and TN tumors are shown in the upper panel (scale bar = 100 *μ*m). Lower panel: human tumor samples (n = number of samples in each subtype) were scored by IHC (see the Materials and Methods section) for LATS1 and LATS2; **P*-value < 0.05, ****P*-value < 0.001. **(H)** Genes were ranked according to fold change between control siRNA-transfected and *LATS1* siRNA-transfected (siLATS1) MCF7 cells and compared with the indicated gene sets (Creighton et al, 2008; Massarweh et al, 2008), using GSEA. **(I)** WT-PyMT, *Lats1*-CKO PyMT, and *Lats2*-CKO PyMT mice were injected IP with 4-hydroxytamoxifen (Tamoxifen) or left untreated (no treatment); n = 12, 18, 7, 10, 9, and 9, respectively.

hyperactivated upon *Lats2* depletion in both mammary tumors (ingenuity activation z-score 0.64; *P*-value of overlap $3.6 \times 10^{-8}$) and mouse livers (Aylon et al, 2016). This resonates with our previous finding that in the liver, LATS2 (but not LATS1) interacts with SREBP1 and its paralog SREBP2 and inhibits their transcriptional activity (Aylon et al, 2016). By similar reasoning, it is plausible that LATS2 and LATS1 may also affect PPARγ activity and cell identity, respectively, in other tissues beyond mammary glands. These effects of LATS2 and LATS1 may be mediated, at least in part, by YAP/TAZ. However, as is often the case for important signaling pathways, multiple components may act in parallel to converge on a common biological outcome. Thus, LATS proteins might execute their functions via a combination of YAP/TAZ-dependent and YAP/TAZ-independent molecular processes. For example, LATS2-mediated phosphorylation of TAZ can indirectly promote PPARγ activity in adipocytes by relieving the inhibitory activity of TAZ on PPARγ (Hong et al, 2005; An et al, 2013), although also within the context of adipogenesis, the upstream Hippo components MST1/2 can promote SAV1–PPARγ complex formation and PPARγ activation (Park et al, 2012). Together with our observations, this implies that multiple Hippo pathway components may modulate PPARγ activity in different biological settings.

Likewise, LATS1 activity may affect cellular identity in contexts beyond mammary tumorigenesis. For instance, liver-specific combined *Lats1* and *Lats2* knockout forces hepatoblasts to commit to biliary epithelial cell lineage (Lee et al, 2016). This occurs via YAP/TAZ repression of *Hnf4a* expression. Furthermore, in preimplantation embryos, LATS-mediated YAP phosphorylation alters cell fate to inhibit the development of the trophectoderm lineage (Nishioka et al, 2009). A similar role in cell fate specification was observed in *Drosophila* for the LATS ortholog *warts* during photoreceptor differentiation (Mikeladze-Dvali et al, 2005). Although our transcriptomic analysis revealed activation of YAP/TAZ transcriptional program preferentially upon *Lats1* deletion, the exact contribution of this activation to the observed phenotypes remains to be elucidated.

Although the robust categorization of breast tumors into subtypes has both molecular and clinical implications, there may exist substantial plasticity in many tumors, confounding unequivocal assignment to a defined subtype (Wahl & Spike, 2017). We observed that deletion of *Lats1* in a lumB-prone breast cancer model results in tumors that exhibit histology and gene expression patterns partially resembling ER-negative, basal-like tumors. Likewise, most of the human BRCA1-associated tumors, typically classified as basal-like, have been shown to originate from luminal progenitors rather than from basal stem cells (Lim et al, 2009; Molyneux et al, 2010). More broadly, the properties of breast cancer propagating cells and the nature of the tumors that they spawn can be shaped extensively by epigenetic and microenvironmental influences (Wahl & Spike, 2017). Of note, reduction in LATS kinase levels in primary human breast epithelial cells increases the number of bipotent and luminal progenitor cells (Britschgi et al, 2017). Interestingly, although we find that depletion of LATS1 in a luminal cancer model enables the increased expression of genes characteristic of a basal-like transcriptional program, Britschgi et al (Britschgi et al, 2017) observed that loss of LATS in basal cells induced the expression of luminal markers. Thus, tumor suppressors such as LATS1 may constrain the inherent plasticity of luminal

progenitor cells, and their loss may augment plasticity in the context of tumorigenesis. Alternatively, it remains possible that in some contexts, mature luminal cells might undergo trans-differentiation, acquiring basal-like characteristics and gene expression patterns (Doherty et al, 2016). Future lineage tracing studies hopefully would resolve this important issue. Either way, we propose that compromised LATS1 expression may be conducive to enhanced tumor cell plasticity, facilitating escape from hormone therapy. Importantly, patients with lumB tumors harboring low levels of *LATS1* might be less responsive to tamoxifen treatment alone and might benefit from more aggressive therapeutic options.

Selective pressure to decrease a particular LATS paralog in breast cancer may depend on distinct subtype-specific signaling events. In particular, LATS proteins were shown to modulate estrogen signaling (Lit et al, 2013; Britschgi et al, 2017) and sustain the canonical functions of the p53 tumor suppressor (Aylon et al, 2006, 2010; Furth et al, 2015). Interestingly, ERα can bind p53 and inhibit p53-dependent transcription (Liu et al, 2006; Konduri et al, 2010). In this scenario, diminished expression of LATS proteins in ER+ breast cancers may have a two-pronged effect in compromising p53 tumor suppressive functionality. In contrast, ER- tumors frequently harbor *TP53* mutations, often resulting in accumulation of mutant p53 proteins (Curtis et al, 2012). Such mutant p53 proteins may acquire oncogenic gain-of-function features (Muller & Vousden, 2014; Shetzer et al, 2016), which can be further augmented by binding to YAP (Di Agostino et al, 2016), whose activity is typically suppressed by the LATS kinases. Altogether, our findings imply that deregulation of LATS1 and LATS2 can exert both common and distinct effects on breast cancer progression, by interaction with a variety of regulatory pathways. Deciphering the nuances of those interactions will be critical for advancing our ability to apply Hippo pathway knowledge towards improving therapeutic options for breast cancer patients.

# Materials and Methods

### Human cancer gene expression and methylation data

The TCGA breast invasive carcinoma gene expression (IlluminaHiSeq log (normalized counts + 1)), methylation (β values from Infinium Methylation 450 k), and clinical data were downloaded from the Xena cancer genome browser (http://xena.ucsc.edu).

A cutoff of the 20% of tumors expressing the lowest levels of each LATS gene was used to divide the tumors into groups (i.e., high versus low expression of each gene). Differences between the four groups of tumors were examined by ANOVA.

LATS1- and LATS2-associated gene signatures were generated based on the above described comparison. The most down-regulated genes in *LATS2*[L] (compared with LATS[H]) tumors were extracted (foldchange > 2, adj*P*-value < 0.05). The 20 genes most differentially down-regulated exclusively in the *LATS2*[L] group of tumors (i.e., not significantly down-regulated in the *LATS1*[L] or LATS[L] groups) were used. The same process was used to generate the LATS1-associated gene signature (Table S4).

For LATS2 promoter methylation analysis, lumB tumors were divided according to LATS2 expression (median cutoff).

Xena cancer genome browser was used to visualize expression data from the TCGA Breast cancer (BRCA) dataset. Tumors were clustered according to PAM50 subtype (PAM50Call_RNAseq). Only tumors with classification were included.

Survival plots were generated either by KM plotter (LATS2 probe 227013_at, Győrffy et al, 2010) or by the R packages "survival" and "survminer," based on the data available in the METABRIC dataset (Curtis et al, 2012). Tumors were divided according to expression (median cutoff). When comparing groups of genes as a signature, z-scores of the expression were calculated per each gene, and the average of these z-scores per sample was used as the signature.

Correlation between methylation and expression was calculated by Pearson's correlation coefficient.

### Animals

All mouse experiments were approved by the Institutional Animal Care and Use Committee (IACUC) of the Weizmann Institute (approval 14521114-1). Genotyping was used to classify littermates into the different experimental groups, and all comparisons of the different genotypes were done between littermates. After sacrificing, the mice were weighed, and tumors were extracted, weighed, and measured. Three tumors consisting of the largest, smallest, and average "representative" tumors were processed further for sample. For analysis of tamoxifen sensitivity, 6-8-wk-old females were injected IP with 20 mg/kg tamoxifen free base (#T5648; Sigma-Aldrich) every other day for 14 d, until the end of the experiment.

### Human tumor samples

Collection of the clinical samples and their experimental use were approved by the Bioethics Committee of the Medical School of Athens, in accordance with the Declaration of Helsinki and local laws and regulations, following also written patient consent. None of the patients received any cancer therapy before surgical resection of the lesions.

### Histopathological evaluation

Histological assessment was performed by three experienced pathologists (ISP, VV, and VGG) based on previously published criteria (Cardiff et al, 2000; Lin et al, 2003; Rudmann et al, 2012).

### Immunohistochemistry

Immunohistochemistry analysis was performed on paraffin-embedded tissues using the primary antibodies listed in Table S5. Heat-mediated antigen retrieval was performed in 10 mM citric acid (pH 6.0). The UltraVision Quanto Detection System was used (#TL-060-QHD; Thermo Fisher Scientific, Bioanalytica) according to the manufacturer's instructions. Hematoxylin was used as counterstain. Evaluation of γH2AX was performed as previously described (Tarcic et al, 2016). For cleaved caspase 3 immunostaining, the % of positive cells was calculated. ERα was evaluated by calculating the % of cancer cells exhibiting intense nuclear staining. For cytokeratin 14, we counted the % of cancer cells exhibiting intense staining at areas

with invasive carcinoma, to avoid counting myoepithelial cells. Evaluation of LATS1 and LATS2 in human breast sporadic carcinomas was based on the method described by Xu et al (2015), using a mixed score (intensity score > 0: negative; 1: weak; 2: moderate; 3: strong and proportion score > 0: negative; 1: < 10%; 2: 11–50%, 3: 51–80%; 4: >80%). In all cases, three independent observers carried out slide examination, with minimal interobserver variability. When analyzed for specific markers, tumors from the same histological subtype were tallied and statistically analyzed, to prevent confounding data due to different tumor spectra in different subtypes.

### Isolation of total RNA, reverse transcription, and RTqPCR

RNA was isolated using the NucleoSpin kit (Macherey Nagel), RNeasy Mini kit (QIAGEN), or RNeasy Microkit (QIAGEN). 1-2 μg of each RNA sample was reverse transcribed using Moloney murine leukemia virus reverse transcriptase (Promega) and random hexamer primers (Applied Biosystems). Real-time qPCR was performed using SYBR Green PCR Supermix (Invitrogen) with a StepOne real-time PCR instrument (Applied Biosystems). For each gene, values for the standard curve were measured and the relative quantity was normalized to *HPRT* or *GAPDH* mRNA.

### Library preparation, RNA-seq, and analysis

#### PyMT tumors
RNA was isolated from *Lats1*-CKO and *Lats2*-CKO PyMT tumors. For each genotype, the corresponding WT littermate controls were used. For RNA-seq analysis, 500 ng of total RNA was processed using the TruSeq RNA Sample Preparation Kit v2 protocol (Illumina) (Part #15026495). Libraries were evaluated by Qubit and TapeStation. Sequencing libraries were constructed with barcodes to allow multiplexing of seven samples per lane (three WT tumors and four *Lats*-CKO tumors). Samples were sequenced on an Illumina HiSeq 2500 V4 instrument. 29–38 million and 35–35 million single-end 60-bp reads were sequenced for the *Lats1*-CKO and the *Lats2*-CKO experiments, respectively.

Reads were mapped using STAR (Dobin and Gingeras, 2015) to mm10 assembly and quantified using mm10 RefSeq annotation. Differential expression analysis was performed on genes with a sum value of at least five counts in all samples using DESeq2 (Love et al, 2014). Raw *P*-values were adjusted for multiple testing using the procedure of Benjamini and Hochberg.

Expression values of differentially expressed genes (rld-log normalized counts) were used to generate hierarchical clustering heatmaps (Pearson's correlation and complete linkage using Partek Genomics Suite). PCA plots of RNA-seq data were generated using rld values with R ggplot.

#### Human cell lines
RNA from ZR75-1 and MCF7 cells transfected with siControl, siLATS1, or siLATS2 oligonucleotides was isolated and subjected to RNA-seq as above. Three and two independent biological replicates were used for ZR75-1 and MCF7 cells, respectively. ZR75-1 cells were transfected with either GFP only, GFP-LATS1, or GFP-LATS2. GFP-positive cells were sorted by FACS 24 h following transfection

(~80,000 cells per sample). Two independent replicates were used. For RNA-seq analysis, 500 ng of total RNA was fragmented, followed by reverse transcription and second strand cDNA synthesis. The double-stranded cDNA was subjected to end repair, A base addition, adapter ligation, and PCR amplification to create libraries. Libraries were evaluated by Qubit and TapeStation. Sequencing libraries were constructed with barcodes to allow multiplexing of multiple samples over three lanes. ~15 million single-end 60-bp reads were sequenced per sample on Illumina HiSeq 2500 V4 instrument.

Reads were trimmed using cutadapt (Martin, 2011) and mapped to assembly hg38. Counting proceeded over genes annotated in RefSeq release hg38 using STAR (Dobin and Gingeras, 2015). Differential expression analysis was performed using DESeq2 (Love et al, 2014) with the cooksCutoff = FALSE, independentFiltering = FALSE set to False. Raw $P$-values were adjusted for multiple testing using the procedure of Benjamini and Hochberg.

The data have been deposited in NCBI's Gene Expression Omnibus (GEO) and are available through GEO series accession number GSE116818.

## Cell lines, transfections, and treatments

All cell lines were maintained at 37°C with 5% $CO_2$. ZR75-1 cells were cultured in RPMI 1640 supplemented with 10% FBS and 1% penicillin + streptomycin (P/S). MDA-MB-468 and MCF7 cells were cultured in DMEM supplemented with 10% FBS and 1% P/S. PyMT-derived cell lines were generated from freshly minced tissue after digestion with 10 mg collagenase A (Roche 10103586001) and 1.5 mg hyluronidase (H4272-30MG; Sigma-Aldrich). After dissociation, the cells were filtered through 70-$\mu$m strainers; washed with DMEM, Hanks' Balanced Salt Solution [+Ca +Mg] supplemented with 2% FBS and 2% Hepes, and Hanks' solution with 8.29 g $NH_4Cl$/l Tris (pH 7.2); and finally resuspended in DMEM. Epithelial cell pellets were obtained by twice centrifuging at 800 $g$ for 5 s. The following day, adherent cells were washed aggressively to detach fibroblasts. Initially, the cells were cultured in DMEM supplemented with 15% FBS, 2 mM glutamine, 1× nonessential amino acids, and 1% P/S. After the cultures stabilized, they were acclimated to and propagated in DMEM supplemented with 10% FBS and 1% P/S.

For siRNA-mediated knockdown, the indicated SMARTpools (Dharmacon, see Table S5) were used with Dharmafect #1 transfection reagent according to the manufacturer's instructions. Final siRNA concentration was 25 nM in all cases. Plasmid transfections were performed using jetPRIME DNA transfection reagent (Polyplus Transfection) according to the manufacturer's instructions. A list of plasmids used can be found in Table S5. Retroviral packaging was performed by jetPEI-mediated transfection of HEK293T or HEK293T Pheonix cells with the appropriate plasmids together with pMD2.G DNA encoding VSV-G envelopes proteins (when infecting human cell lines). Virus-containing supernatants were collected 48 h following transfection, filtered, and supplemented with 8 $\mu$g/ml polybrene. Infected ZR75-1 and PyMT cells were selected with 1.5 or 1 $\mu$g/ml blasticidin, respectively. The cells were treated with 1 $\mu$M or 5 $\mu$M 5-aza-2'deoxycytidine each day for four consecutive days. Glucose starvation and treatments with RGZ were conducted in media containing 1% serum.

## Western blots

Cell pellets were resuspended in protein sample buffer and boiled. The samples were resolved by SDSPAGE. Imaging was performed using a ChemiDoc MP imaging system (Bio-Rad) with the Image Lab 4.1 program (Bio-Rad).

## Imaging flow cytometry (ImageStream)

Cells were collected with trypsin, washed with PBS, and fixed in 3.5% PFA followed by permeabilization with 0.1% Triton. Washes were done in PBS supplemented with 1% FCS and 2 mM EDTA. Cells were incubated with the indicated primary antibody for 1 h at room temperature, followed by washes and 45 min of incubation with fluorescent-conjugated secondary antibody (GaR Alexa 647, #A21244; LifeTech or GaM Alexa 595, #A11032; LifeTech) and DAPI (#D1306; LifeTech). The cells were imaged by ImageStreamX mark II (Amnis, part of EMD Millipore) using bright field 488 nm, 561 nm, and 642 nm lasers. At least 30,000 cells were collected from each sample and data were analyzed using image analysis software (IDEAS 6.2; Amnis Corporation). Images were compensated for fluorescent dye overlap by using single-stain controls. Gating was done for single cells, using the area and aspect ratio features, and for focused cells using the gradient RMS feature, as previously described (George et al, 2006). Data were analyzed with the IDEAS 6.1 software (Amnis, part of EMD Millipore). Only cells with intact nucleus (according to DAPI staining) were analyzed. Nuclear localization was determined by the similarity feature on the nuclear mask of the DAPI staining and the GFP signal (the log transformed Pearson's correlation coefficient in the two input images). PPARγ-positive cells were gated on the basis of comparison with nonstained sample.

## PI exclusion assay

Cells, including floaters, were collected using trypsin, washed once with PBS, and resuspended in PBS containing 12.5 $\mu$g/ml PI. Percentage of dead cells (positive for PI staining) was measured for each sample using Guava EasyCyte Flow Cytometer (Merck Millipore).

## Measurement of oxygen consumption rate (OCR) and extracellular acidification rates (ECAR) (Seahorse analysis)

OCR and ECAR assays were performed according to the manufacturer's instructions using the XFe96 or XFp Extracellular Flux analyzers (Agilent Technologies).

ZR75-1 and PyMT cells were plated at 20,000 and 5,000 cells in 80 $\mu$l per well, respectively, in Agilent Seahorse cell culture microplates (Cat No. 101085-004). The cells were incubated for 48 h in a humidified 37°C incubator with 5% $CO_2$ in RPMI (ZR75-1 cells) or DMEM (PyMT cells) media supplemented with 1% FCS. One hour before performing the assay, growth medium in the wells of an XF cell plate was exchanged with the appropriate assay medium (pH 7.4 ± 0.1 at 37°C) by three consecutive washes leaving 180 $\mu$l of medium per well in the end. For ECAR measurements, XF base medium 4 (Agilent Technologies) supplemented with 2 mM L-glutamine and for OCR measurements, 2 mM L-glutamine, 1 mM sodium pyruvate, and 10 mM D+ glucose. The cells were incubated at 37°C (no $CO_2$) for 60 min to allow temperature and pH

equilibration. Meanwhile, hydrated XFe96/XFp sensor cartridge ports were loaded with respective assay compounds diluted in the assay medium. For ECAR, four injection cycles consisted of: 10 mM glucose (#G7528; Sigma-Aldrich) 1 $\mu$M oligomycin (#O4876; Sigma-Aldrich) and then 50 mM 2-DG (#D6134; Sigma-Aldrich). For OCR measurements, injection cycles were as follows: 1 $\mu$M oligomycin, 1 $\mu$M FCCP (#C2920; Sigma-Aldrich), and 0.5 $\mu$M rotenone/antimycin A (#R8875/#A8674; Sigma-Aldrich) were injected consecutively. Each cycle consisted of 3-min mixing and 3-min measurement. The average rates of each cycle were used for computations.

Nuclear stain with CyQuant (#C35011; Thermo Fisher Scientific) was used for normalization. Basal respiration was calculated by subtracting the nonmitochondrial respiration rate (minimum rate measurement after rotenone/antimycin A addition) from the last measurement rate before oligomycin addition. Basal glycolysis was calculated by subtracting the last rate measurement before glucose addition from the maximum rate measurement after glucose addition.

### Functional analysis of gene expression data

For GSEA analysis (Subramanian et al, 2005), genes were ranked according to fold change between the two described conditions, with only significant differences considered (PyMT CKO experiments *P*-value < 0.05; siRNA in human cell lines adj*P*-value < 0.05; over-expression in human cell lines average FC > 1.5). Comparison with different gene sets (Table S6) was done using GSEA preranked tool. Similar ranking was used to analyze gene expression patterns by Ingenuity Pathway Analysis software (QIAGEN Inc., https://www.qiagenbioinformatics.com/products/ingenuity-pathway-analysis/). KEGG pathways and GO-BP enrichment analysis was performed using DAVID (Huang et al, 2008). Pathway enrichment analysis was also performed using GeneAnalytics (Ben-Ari Fuchs et al, 2016).

To derive a basal-associated gene signature, tumors in the BRCA dataset of TCGA were divided according to tumor type, and the genes most differentially expressed between luminal and basal tumors were extracted (adj*P*-value < 0.05, FC > 1.5) and analyzed for enriched biological pathways (Metascape). Both differentially expressed genes and genes associated with the most differential pathways were compared with a list of differentially expressed genes in either *Lats1*- or *Lats2*-CKO PyMT tumors (each compared with its littermate WT-PyMT controls). All overlapping genes were used to define the signature.

### Statistics

Unless noted otherwise, *P*-values were determined using two-tailed *t* tests (paired or two-sample, depending on experimental setting).

## Supplementary Information

## Acknowledgements

We thank Giuseppe Mallel for fruitful discussions and valuable inputs; Sima Benjamin, Dana Robbins, Orit Zion, and Shlomit Gilad (Israel National Center for Personalized Medicine, Weizmann Institute of Science [WIS]) for their help with RNA-seq; Gil Stelzer (Life Sciences Core Facilities, WIS) for assisting with pipeline analysis of human cell line RNA-seq results; Ori Brenner (Veterinary Resources, WIS) for help with histological analysis; and Ziv Porath (Life Sciences Core Facilities, Faculty of Biochemistry, WIS) for ImageStream data acquisition and analysis. We thank James Martin (Baylor College of Medicine, Houston, TX, USA) for the generous sharing of conditional knockout mice. This work was supported by grants from the Rising Tide Foundation, the Dr. Miriam and Sheldon G. Adelson Medical Research Foundation, a Center of Excellence grant from the Israel Science Foundation, the Comisaroff Family Trust, the Estate of John Hunter, and the Moross Integrated Cancer Center. M Oren is incumbent of the Andre Lwoff chair in molecular biology.

## Author Contributions

N Furth: conceptualization, formal analysis, funding acquisition, validation, investigation, visualization, methodology, and writing—original draft, review, and editing.
IS Pateras: formal analysis and investigation.
R Rotkopf: formal analysis, investigation, and methodology.
V Vlachou: formal analysis and investigation.
I Rivkin: investigation.
I Schmitt: investigation.
D Bakaev: investigation.
A Gershoni: investigation.
E Ainbinder: investigation.
D Leshkowitz: data curation and formal analysis.
RL Johnson: resources.
VG Gorgoulis: supervision, funding acquisition, and investigation.
M Oren: conceptualization, supervision, funding acquisition, methodology, and writing—review and editing.
Y Aylon: conceptualization, formal analysis, supervision, funding acquisition, validation, investigation, visualization, methodology, and writing—original draft, review, and editing.

## Conflict of Interest Statement

The authors declare that they have no conflict of interest.

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
