## [Reviewer comments · Life Science Alliance]

Life Science Alliance

LATS1 and LATS2 suppress breast cancer progression by maintaining cell identity and metabolic state

Noa Furth, Ioannis Pateras, Ron Rotkopf, Vassiliki Vlachou, Irina Rivkin, Ina Schmitt, Deborah Bakaev, Anat Gershoni, Elena Ainbinder, Dena Leshkowitz, Randy Johnson, Vassilis G. Gorgoulis, Moshe Oren, and Yael Aylon

DOI: [10.26508/lsa.201800171](https://doi.org/10.26508/lsa.201800171)

Corresponding author(s): Yael Aylon, The Weizmann Institute of Science and Moshe Oren, The Weizmann Inst. of Science

Review Timeline:

Submission Date:	2018-08-23
Editorial Decision:	2018-08-23
Revision Received:	2018-09-07
Editorial Decision:	2018-09-14
Revision Received:	2018-10-09
Editorial Decision:	2018-10-12
Revision Received:	2018-10-13
Accepted:	2018-10-15

Scientific Editor: Andrea Leibfried

Transaction Report:

Please note that the manuscript was previously reviewed at another journal and the reports were taken into account in the decision-making process at Life Science Alliance. Since the original reviews are not subject to Life Science Alliance's transparent review process policy, the reports and author response cannot be published.

August 23, 2018

Re: Life Science Alliance manuscript #LSA-2018-00171-T

Dr. Yael Aylon
The Weizmann Institute of Science
Department of Molecular Cell Biology
Rehovot
Israel

Dear Dr. Aylon,

Thank you for transferring your manuscript entitled "Enhanced breast cancer progression upon loss of LATS1 or LATS2 reveals differential impact on metabolic and differentiation states" to Life Science Alliance. The manuscript was assessed by expert reviewers at another journal before, and the journal editors have transferred those reports to us with your permission.

The reviewers who assessed your work at the other journal noted that many of your observations are interesting, and the main criticism of the reviewers pertained to the level of definitive insight offered at this stage. Based on these reviewer reports already at hand, we would like to invite you to submit a revised version for publication in Life Science Alliance.

While there are numerous points that were raised by the reviewers, most concerns can be addressed by re-wording, toning down the conclusions, and further discussion to allow publication here. We'd furthermore appreciate if a revision addresses:

- the minor issues mentioned by reviewer #1
- point 3 of reviewer #2 (performing a more detailed analysis of the data already at hand)
- point 4 of reviewer #3

To upload a point-by-point response and the revised version of your manuscript, please log in to your account: <https://lsa.msubmit.net/cgi-bin/main.plex>
You will be guided to complete the submission of your revised manuscript and to fill in all necessary information.

Thank you for this interesting contribution to Life Science Alliance. We are looking forward to receiving your revised manuscript.

Sincerely,

- A letter addressing the reviewers' comments point by point.
- An editable version of the final text (.DOC or .DOCX) is needed for copyediting (no PDFs).
- High-resolution figure, supplementary figure and video files uploaded as individual files: See our detailed guidelines for preparing your production-ready images, <http://life-science-alliance.org/authorguide>
- Summary blurb (enter in submission system): A short text summarizing in a single sentence the study (max. 200 characters including spaces). This text is used in conjunction with the titles of papers, hence should be informative and complementary to the title and running title. It should describe the context and significance of the findings for a general readership; it should be written in the present tense and refer to the work in the third person. Author names should not be mentioned.

B. MANUSCRIPT ORGANIZATION AND FORMATTING:

Full guidelines are available on our Instructions for Authors page, <http://life-science-alliance.org/authorguide>

September 14, 2018

Re: Life Science Alliance manuscript #LSA-2018-00171-TR

Dr. Yael Aylon
The Weizmann Institute of Science
Department of Molecular Cell Biology
234 Herzl Street, POB 26
Rehovot 7610001
Israel

Dear Dr. Aylon,

Thank you for submitting your revised manuscript entitled "LATS1 and LATS2 suppress breast cancer progression by maintaining cell identity and metabolic state" to Life Science Alliance.

We assessed your work and the introduced changes, and while we are still supportive of publication of your work in Life Science Alliance, we would like to suggest to include further clarifications / changes of data representation.

Importantly, we'd appreciate if you could send us a full point-by-point response to the reviewer comments you've received at the other journal. Furthermore, we'd appreciate that a further revision includes the following:

- change the data representation of the PI exclusion assay (reviewer #1)
- comment on / explain the observed latency of the MMTV PyMT control mice (reviewer #2) and the possibility that the background rather than LATS deficiency leads to the observed effects
- comment on / explain the transcription pattern resemblance of Lats2-CKO PyMT adenoma / MIN and WT-PyMT carcinoma (reviewer #2)
- explain why lower but not higher concentration of 5-Aza have better effect on elevating Lats2 expression level in Fig 3A (reviewer #3)

Thank you for this interesting contribution to Life Science Alliance. We are looking forward to receiving your revised manuscript.

Sincerely,

Andrea Leibfried, PhD
Executive Editor
Life Science Alliance
Meyerhofstr. 1
69117 Heidelberg, Germany

t +49 6221 8891 502
e a.leibfried@life-science-alliance.org
www.life-science-alliance.org

- A letter addressing the reviewers' comments point by point.
- An editable version of the final text (.DOC or .DOCX) is needed for copyediting (no PDFs).
- High-resolution figure, supplementary figure and video files uploaded as individual files: See our detailed guidelines for preparing your production-ready images, <http://life-science-alliance.org/authorguide>
- Summary blurb (enter in submission system): A short text summarizing in a single sentence the study (max. 200 characters including spaces). This text is used in conjunction with the titles of papers, hence should be informative and complementary to the title and running title. It should describe the context and significance of the findings for a general readership; it should be written in the present tense and refer to the work in the third person. Author names should not be mentioned.

B. MANUSCRIPT ORGANIZATION AND FORMATTING:

Full guidelines are available on our Instructions for Authors page, <http://life-science-alliance.org/authorguide>

October 12, 2018

RE: Life Science Alliance Manuscript #LSA-2018-00171-TRR

Dr. Yael Aylon
The Weizmann Institute of Science
Department of Molecular Cell Biology
234 Herzl Street, POB 26
Rehovot 7610001
Israel

Dear Dr. Aylon,

Thank you for submitting your revised manuscript entitled "LATS1 and LATS2 suppress breast cancer progression by maintaining cell identity and metabolic state". We appreciate your point-by-point response made to the criticisms received during peer review elsewhere and the changes introduced into the manuscript. We would thus be happy to publish your paper in Life Science Alliance pending final revisions necessary to meet our formatting guidelines:

- please arrange FigS6 on a single page (currently split into two pages)
- please add a callout in the manuscript text for Table S6.

A. FINAL FILES:

-- High-resolution figure, supplementary figure and video files uploaded as individual files: See our detailed guidelines for preparing your production-ready images, <http://life-science-alliance.org/authorguide>

B. MANUSCRIPT ORGANIZATION AND FORMATTING:

Full guidelines are available on our Instructions for Authors page, <http://life-science-alliance.org/authorguide>

Sincerely,
